# Position: Assistive AI Requires Personalized Specialists, not Generalists

**Homanga Bharadhwaj** [1]

## Abstract

The AI community is rapidly converging on *generalist* foundation models trained on web-scale data. While this paradigm has yielded impressive gains, we argue that reliable assistive AI requires a complementary objective: systems that become increasingly well matched to a particular individual, local environment, and interaction history. The most valuable assistants will not merely be those that can attempt many tasks for many users, but those that can do the right things for a specific person over a long period of time. We take the position that *deployment-adaptive specialists* are a central target for high-impact assistive AI: systems whose specialization is defined not by narrow task taxonomies, but by sustained coupling to an individual user, their local environment, and their evolving interaction history. This is not a rejection of generalist pretraining: broad foundation models provide increasingly strong priors. The unresolved scientific problem is converting those priors into stable local competence without repeated costly mistakes. We substantiate this argument through three case studies: (i) AI agents that help humans automate web activities; (ii) wearable assistants that predict actions in-context from continuous egocentric streams; and (iii) home robots that assist in daily tasks under safety and compliance constraints. In these settings, the most critical data is often generated *after deployment* as a streaming, privacy-sensitive, on-policy interaction stream. We outline research directions for building locally adaptive assistants that learn from organic observational data, avoid self-reinforcing errors, and improve safely over long horizons.

[1]Department of Computer Science, Johns Hopkins University. Correspondence to: Homanga Bharadhwaj <hbharad2@jhu.edu>.

*Proceedings of the $43^{rd}$ International Conference on Machine Learning*, Seoul, South Korea. PMLR 306, 2026. Copyright 2026 by the author(s).

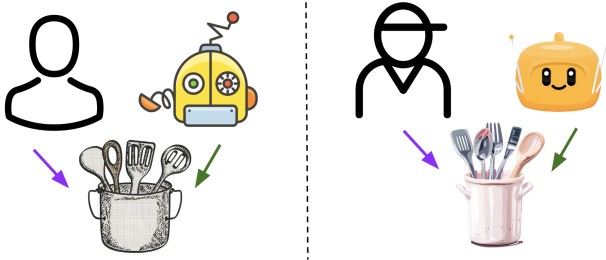

*Figure 1.* Personalized AI assistance is a longitudinal specialization problem. Different robots helping different people in different homes may face similar tasks, such as cooking or cleaning, but need to act very differently depending on local layouts, objects, routines, preferences, and safety constraints. These requirements also keep evolving over time, so effective assistive AI must continually specialize to the particular user and environment it is used in.

## 1. Introduction

> *"Plans are resources for situated action, but do not in any strong sense determine its course."*
>
> — *Lucy Suchman (1987)*

Building generalist systems capable of zero-shot performance in diverse tasks has been a grand pursuit in AI, and recent progress suggests a real path to broad competence via large-scale domain-specific datasets and general training recipes that can ingest large datasets (OpenAI, 2024; Kaplan et al., 2020). In research practice, generalist models are appealing because they offer a single artifact that benchmarks well across many tasks, amortizes training cost, and serves as a reusable backbone for downstream adaptation. In product practice, generalist models are attractive because they can be deployed widely and potentially alleviate domain-specific needs. At the same time, leading deployed assistants already expose a partial version of the specialist agenda: they use memory, long contexts, retrieval over prior interactions, custom instructions, and product metrics that ask whether the system is becoming useful for a particular user. These motivations are sound, and they have yielded systems that can write, code, analyze, generate, and converse at a level that was implausible a few years ago (Radford et al., 2021; Maaz et al., 2023; Ho et al., 2022; Yang et al., 2025).

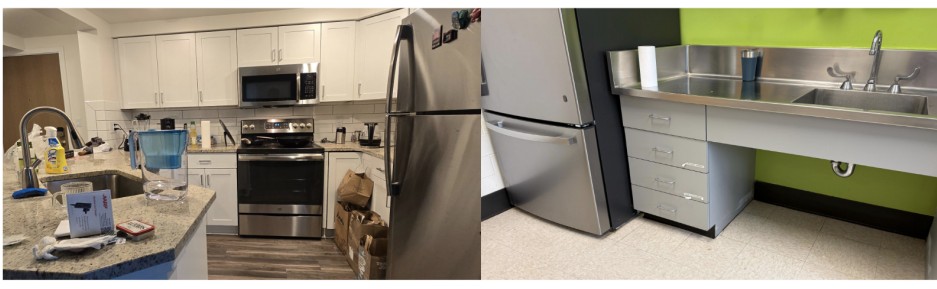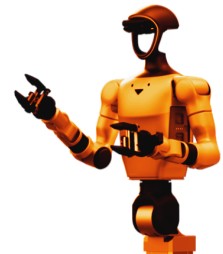

*Different Homes have very different layout / objects / human preferences*

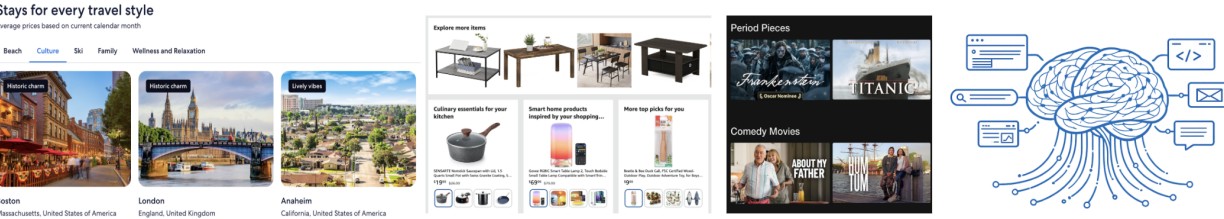

*Different web users have evolving travel / shopping / movie preferences*

*Figure 2.* **Top Row.** Home Robots will need to effortlessly adapt to new homes and evolving user preferences quickly as they are deployed. This requires quick continual adaptation to *specialize* in the local environment. **Bottom Row.** Web Agents similarly need to adapt to different user preferences efficiently for daily activities. As these technologies become ubiquitous, end users would care about the performance of these models in their own specific context, instead of general performance in others' contexts.

However, to make progress towards building AI assistants that can help humans with daily activities, we need to build systems that are reliably useful in the particular *personalized* contexts where people live, work, and care about specific needs. In those contexts, the most important dimensions of "intelligence" are often local rather than universal: how a person likes information presented, what sources they trust, which tradeoffs they prefer, what their home layout looks like, where objects are kept, which routines are sacred, which mistakes are unacceptable, and which norms define "helpful" versus "annoying" (Clark & Chalmers, 1998)

**This position paper argues that the last mile of assistive AI is local specialization**: the field should treat deployment-time adaptation to a user and environment as a primary scientific objective rather than as a secondary product pursuit. This is not a call to return to narrow task-specific ML, nor a claim that generalist pretraining is useless or incapable of further progress. Crucially, *specialization does not mean single-task*. A specialist can be a multi-task system capable of a broad set of tasks, but its competence is concentrated within a *bounded* set of scenarios defined by a particular user and environment. A useful analogy is a home robot that can cook, clean, do laundry, and assist with errands, but does so *in this specific home*: the layout, objects, routines, and norms are stable enough that the relevant variations are limited, learnable, and worth exploiting (Fig. 2). In this sense, specialization is not about shrinking the action space to one function; it is about *tightening the deployment distribution* so that broad capabilities become reliable through

continual exposure and adaptation. The relevant unit of generalization is therefore not "any user, any environment," but *"this user, this environment, across time."*

Our contribution is a reframing rather than the introduction of continual learning, privacy-preserving learning, memory, or adaptation as new subfields. These areas already have deep literatures (Kirkpatrick et al., 2017; Parisi et al., 2019; De Lange et al., 2021; Thrun, 1998), and many deployed assistants already perform some form of user-specific adaptation in context space. The position here is narrower: assistive AI combines these ingredients under a distinctive deployment regime that remains under-emphasized in academic benchmarks and in algorithms for safe post-deployment learning. Local data is privacy-sensitive, feedback is sparse and implicit, the environment is non-stationary, and mistakes are costly because they affect a real person in a real context. Under this regime, the central research question is not only how to build stronger general-purpose priors, but how to make those priors safely specialize after deployment.

This claim is especially evident in embodied AI. When we have personal robots, we will not care if the robot in one home performs equally well in another home. We will care that the robot in *our* homes understands *our* kitchens, *our* clutter patterns, *our* safety constraints, and *our* preferences about how tasks should be done. A robot that can do "everything" in a generic lab setup but fails to adapt to household idiosyncrasies is not a useful product. Conversely, a robot that is mediocre out-of-the-box but rapidly becomes excel-

lent within a single household through continual adaptation may be far more valuable.

The rest of the paper develops this position, formalizing what we mean by specialists, motivating the case via web agents, assistive wearables, and home robots, and outlining the learning and evaluation challenges required to make continual specialization reliable.

## 2. Deployment-Adaptive Specialist Models

In contemporary discourse, the term *specialist* often means "narrow" and *generalist* means "broad." That framing confounds two distinct axes: (i) the breadth of tasks a model can in principle attempt, and (ii) the degree to which the model is *coupled* to a particular user, environment, tool-chain, and feedback stream. In this paper, we use *specialist model* in the second sense. A specialist is a learning system whose behavior is optimized for a specific deployment context, such that its behavior meaningfully depends on local history, local constraints, and local human preferences. A generalist model, by contrast, is optimized primarily for performance averaged across many users and environments, and is often evaluated by its ability to behave plausibly without relying on long-term individual personalization.

This distinction is especially important for assistive AI because success is often not determined by a universal definition of correctness, but by preference, convention, and appropriate reliance. A web agent is "correct" when it surfaces sources a user trusts, formats information the way they prefer, and acts within their desired autonomy boundaries. A home robot is "correct" when it cleans in the way a household considers clean, respects fragile or off-limits items, and learns routines that are rarely written down. Two users can issue the same instruction and mean different things, and these differences are often stable enough to learn but too local, private, and evolving to be captured by generic training data. We therefore view many assistive settings not merely as distribution shift, but as *distribution individuation*: the relevant deployment distribution is shaped by one person, one environment, and their history over time.

Specialization here is not the same as the classical pre-deep-learning notion of an appliance or rigid embedded controller. Microwaves, washing machines, and televisions are specialists only in a fixed functional sense: their behavior is designed in advance. Our claim is about *learning specialization*. A specialist model may perform many tasks, but it becomes useful by learning the quirks and norms of a particular setting. A household robot that learns how *this* home stores dishes, or a web agent that learns *this* user's trusted sources and risk tolerance, is a specialist even if its task list is broad. Conversely, a proficient multi-skill system remains a non-specialist if it does not reliably adapt to the individual or environment.

This should not be read as a binary taxonomy of architectures. A deployed system may combine a generalist backbone with retrieval, long-context memory, editable memory records, compact latent state, adapters, targeted parameter updates, or task-specific skills. What makes it a specialist in our sense is not a particular mechanism, but the fact that the deployed artifact is *non-static*: its behavior becomes meaningfully shaped by local history, constraints, and feedback. A useful implementation blueprint is therefore not "one model per task," but a layered assistive system: (i) a broadly trained foundation-model backbone; (ii) persistent, user-editable memory over preferences, routines, objects, and past corrections; (iii) user- and environment-specific latent state that can be updated quickly; (iv) skill or policy modules adapted within bounded regions of the state-action space; and (v) safety monitors that gate updates, detect drift, and support rollback (Bharadhwaj et al., 2021; Bharadhwaj, 2022; Kang et al., 2020). This distinction matters for trust: durable preferences and memories should often be inspectable and steerable, even when some inference state remains latent (Adomavicius & Tuzhilin, 2005; Koren et al., 2009).

This framing clarifies the gap between current progress and the specialist motivation. Many deployed assistants already use instructions, memories, retrieval, long contexts, and personalization. These are real forms of adaptation, but academic benchmarks and model releases still disproportionately reward static, cross-user performance rather than safe, measurable improvement for a particular user after deployment. The hard problem is building systems that become *our* web agent, *our* wearable assistant, and *our* home robot over long horizons.

## 3. Case Study I: AI Web Agents

Capable AI agents that can aid humans with tasks on the web need to navigate complex interfaces, maintain memory over time, interpret ambiguous instructions, and recover from errors (Ning et al., 2025; Hu et al., 2025; Deng et al., 2023). But what determines their usefulness is not generic browsing skills but alignment to the end user's idiosyncratic preferences: which sources they trust, how they trade off recency vs. credibility, whether they want concise answers or thorough syntheses, how they prefer citations and provenance, and how risk-tolerant they are about taking actions (e.g., airline booking, purchasing items on e-commerce platforms, form-filling). Users also differ in when they want clarification vs. autonomy, and these preferences vary with context and stakes, echoing classic mixed-initiative design questions about when a system should act, ask, explain, or defer (Horvitz, 1999; Amershi et al., 2019).

In practice, many agents already use prompt-time instructions, persistent memories, custom settings, and retrieved interaction history to adapt in context. This is an important and genuinely useful form of specialization, but it remains brittle when preferences are implicit, drifting, or entangled with high-stakes actions. A specialist web agent should therefore infer preferences and task regularities from interaction history, store durable ones in interpretable forms, and update them continually using weak feedback: acceptance/rejection, corrections, click patterns, ignored results, and user edits. This is the mechanism that turns broad competence into reliable assistance.

The web itself is also nonstationary. Interfaces update, flows change, and the long tail of sites contains idiosyncratic patterns not captured by static training. A specialist benefits from continual learning in the user's actual browsing environment: learning stable navigation sequences for an employer portal, extracting the right fields from a preferred travel site's quirky UI, or adapting to recurring page templates of the user's trusted sources. The key is to learn both the content of an update and its appropriate privacy scope. The most informative data for a web agent is often generated *after deployment*, and progress depends on learning from that on-policy stream under real constraints.

## 4. Case Study II: In-Context Predictions via Wearables

A second, increasingly critical domain for specialization is *in-context action prediction* enabled by the proliferation of wearable devices—smart glasses, watches, and emerging AR platforms (Engel et al., 2023; Guzov et al., 2025; Chen et al., 2025). Unlike web agents constrained to browser environments, these systems operate within the physical world, observing and potentially assisting with daily activities ranging from cooking new dishes and assembling furniture to memory support and just-in-time guidance. Concretely, a wearable assistant should be able to answer questions like *"what did I eat earlier today?"*, *"where did I leave my phone in the house?"*, or *"what did I promise to do next?"* by retrieving and grounding relevant moments from long personal histories, and it should also anticipate what the user is likely to do next—preemptively offering reminders, safety checks, or lightweight suggestions before errors occur. These capabilities require two core competencies: robust future anticipation from partial observations and reliable retrieval over long, noisy, multimodal contexts.

This domain is intrinsically personalized. Even when executing identical tasks, such as following a recipe, users exhibit distinct behaviors regarding workspace layout, tool selection, pacing, and safety norms. Preferences also govern what "help" means: a novice may want detailed step-by-step coaching and frequent confirmation, whereas an expert may prefer subtle, low-interruption prompts or silent logging for later recall. The same intervention can therefore be helpful for one user and actively harmful (annoying, distracting, or unsafe) for another, making the wearable assistant's policy fundamentally preference-coupled.

Wearables generate dense, continuous multimodal streams such as egocentric video, audio, and increasingly physiological signals that are temporally correlated and highly sensitive. The fundamental challenge lies in transforming this organic, on-policy data into reliable assistance under strict privacy and usability constraints. This involves inferring latent intent, detecting mistakes or unsafe deviations before they escalate, and converting weak feedback signals into continual improvement. Thus, the highest-value learning occurs *after deployment* and *on-device*: the system becomes tightly coupled to an individual's embodied reality, building long-horizon memory of routines and object locations, learning when to speak versus stay quiet, and improving over time without repeatedly re-learning the same user-specific norms.

## 5. Case Study III: Home Robots

Embodied AI makes the case for specialization especially tangible: homes are messy, safety-critical, and full of long-tail variation, yet the home robot's actual deployment environment is usually one specific household. From an end-user perspective, the robot does not need to be equally good in every home; it needs to get reliably better in the home it is deployed in, and to keep improving over time as the household changes. This shifts the goal from global generalization to *within-home* generalization across rooms, lighting, clutter, seasons, and routines, while maintaining robust, reliable behavior.

However, much of the current robot learning advances are optimized for a different goal: the *zero-shot* setting. A large fraction of recent progress aims to train a fixed policy (often at scale) that performs immediately in unseen environments without post-deployment learning. This includes large-scale behavior cloning from broad teleoperation datasets (Brohan et al., 2022; Zitkovich et al., 2023; Kim et al., 2024; Black et al., 2025), data generation in simulation (NVIDIA et al., 2025; Mandlekar et al., 2023; Zhou et al., 2025), and approaches that leverage human videos as supervision (Bharadhwaj et al., 2024; Bahl et al., 2023; Jain et al., 2024; Kareer et al., 2025; Wang et al., 2023; Shaw et al., 2022). These efforts are valuable as priors, and generalist robot policies may continue to improve rapidly. Our point is narrower: even strong zero-shot competence leaves unresolved how a deployed robot should become reliably aligned to one home, one set of routines, and one household's safety constraints over time.

The reason this deployment problem remains difficult is that

robotics is hard in ways that make purely static evaluation brittle and real-world household deployment challenging. Generalization breaks on exactly the phenomena that define homes: clutter that changes daily, contact-rich manipulation, deformables, reflective and transparent surfaces, occlusions, sensing artifacts, tool variation, hardware drift, and under-specified user preferences. Minor distribution shifts—new lighting, a rearranged drawer, a different brand of packaging—can cascade into failure because perception, state estimation, planning, and control are tightly coupled. This zero-shot generalization is not just challenging, it is systematically undercut by the long tail of physical interaction and the cost of errors.

This motivates the specialist view: rather than treating the diversity of homes as a single distribution to be covered by a static, zero-shot policy, we should treat *deployment* as the primary data regime and *adaptation* as the core capability. A household specialist should quickly build a model of *this* home and its people—where objects typically end up, which drawers stick, how lighting and reflections change at night, what "clean" means in this household, and which items are fragile, off-limits, or emotionally important. Much of this knowledge is implicit and idiosyncratic: the decorative towel versus the usable one; the single pan that cannot be scratched; the household's preferred way to load the dishwasher. Repeated exposure to the same space and routines is exactly the personalized structure that makes high reliability achievable.

However, the challenges of exploration and interactive online learning are exacerbated by safety and compliance requirements. Even minor repeated failures quickly erode trust, and in homes they can be genuinely hazardous—broken glass, knives, liquids near electronics, pets and children. A practical path to household robotics competence will likely combine strong priors from large-scale pretraining with personalized adaptation, including simulators in the loop. Thus the central scientific question is not whether a single policy can work everywhere out of the box, but whether a robot can enter a new home and *rapidly become reliable there*, improving from interactive household experience without accumulating unsafe failures and forgetting past experiences.

## 6. From Web-Scale Pretraining to Streaming Specialization

In practice, specialist systems will often be *initialized* from generalists. Web-scale pretraining has been the dominant success story of modern ML, producing powerful, reusable priors—language competence, broad visual representations, and general multimodal abstractions—that reduce sample complexity and provide strong heuristics for perception, instruction-following, and planning (Radford et al., 2019;

OpenAI, 2024; Saharia et al., 2022; Kirillov et al., 2023; Goswami et al., 2024). This makes a "generalist → specialist" pipeline appealing: a shared backbone provides broad competence, while specialization concentrates that competence within a bounded deployment distribution. The paper therefore should not be construed as denying the trend that generalist models often outperform narrow specialists on static tasks; rather, it asks what additional mechanisms are needed when the deployment scenario is longitudinal, local, privacy-sensitive assistance.

However, the framing of the pipeline is also where the hard problems concentrate. In web, wearable, and robotic domains, the deployable backbone may be far smaller than frontier text models because of latency, power, privacy, and local-compute limits, so the prior can be fragile or poorly grounded in constrained embodied and interactive settings. A strong prior still does not tell us how the system should update from endogenous streams; track evolving preferences; avoid self-reinforcing errors under feedback loops; or remain stable under non-stationarity. Specialization can therefore act as a compute-efficient equalizer: not a substitute for better generalists, but a way to convert imperfect priors into local reliability.

The key difference is not that specialists have *more* data, but that they have a different *data source and lifecycle*. The most relevant datapoints for a specialist often arrive *after deployment* as a continuous stream from one or few users in one or few environments, where the system's own behavior influences what it observes. These deployment streams violate the core assumptions of typical foundation model training: they are non-i.i.d. and frequently on-policy, temporally correlated, shaped by the agent's actions, and entangled with feedback loops (what the agent chooses affects what it learns next). They are also non-stationary: user preferences drift, websites update, and homes evolve. Finally, the highest-value data streams are often privacy-sensitive and difficult to centralize, making locality constraints (on-device learning, secure data repositories, privacy-preserving aggregation) part of the learning problem rather than an infrastructure detail.

A specialist-centric view therefore emphasizes the *learning loop* over the pretraining corpus. The system must decide what experiences to acquire, how to interpret weak feedback, and how to incorporate new information without overwriting previously acquired learning. This is aligned with recent test-time learning work, which asks whether training can equip models to keep adapting on a particular problem rather than only perform fixed-parameter inference (Yuksekgonul et al., 2026). It foregrounds two requirements: (i) continual learning from temporally correlated streams under feedback loops and drift, and (ii) fast adaptation from sparse, implicit signals with minimal user burden and minimal risk.

This suggests a layered deployed system: a generalist backbone paired with persistent memory, environment-specific skills, local latent state, and safety monitors. Specialization need not update all weights; it can occur through targeted updates, structured memory, or latent variables that capture local regularities while preserving priors. Recent LLM agents already move in this direction: AWS DevOps Agent's learned skills turn operational traces into structured knowledge files, and Claude Dreams curates agent memory across sessions (Amazon Web Services; Anthropic). Making this explicit, measurable, and safe remains open. Thus, web-scale pretraining can bootstrap assistive AI, but streaming specialization is where reliability will ultimately matter.

## 7. Research Challenge I: Continual Learning from Streaming Data

For deployment-adaptive specialists, continual learning is a key requirement: systems ingest temporally correlated, on-policy streams whose content is shaped by the agent's own decisions. This connects to lifelong learning, catastrophic forgetting, online adaptation, and long-term embodied autonomy (Thrun, 1998; Kirkpatrick et al., 2017; Parisi et al., 2019; De Lange et al., 2021; Kunze et al., 2018). What differs in assistive deployment is the coupling of constraints: the data is personal and difficult to pool, the distribution drifts as users and environments change, and the system's actions affect which evidence it receives. These properties move beyond the common task-sequence formulation of continual learning and closer to closed-loop online learning, where the learner's policy changes the future data distribution (Ross et al., 2011; De Lange et al., 2021). For web agents this includes UI states, tool traces, and user edits; for robotics it includes egocentric perception, proprioception, robot-object contacts, and human interventions. Wearables intensify these constraints through continuous multimodal streams such as video, audio, and physiological signals, where useful supervision is weak, implicit, and often too sensitive to centralize (Bao et al., 2025; Engel et al., 2023; Guzov et al., 2025; Chen et al., 2025). The core difficulty is deciding what to retain, what to revise, and what to ignore while the world and user drift.

A key constraint is that streaming specialization creates feedback loops. If a web agent commits early to a slightly wrong workflow, source preference, or vendor choice, it may stop visiting the states that would correct it, entrenching errors through self-confirmation. If a household robot avoids uncertain cabinet, drawer, or deformable-object interactions, it may never collect the contact-rich evidence required to improve; if it explores aggressively, it may damage objects or violate household norms. Wearables face an analogous problem: frequent interventions may bias user behavior, while overly passive behavior may miss moments that reveal intent, routine, or safety-relevant deviations. This pushes continual learning toward closed-loop formulations where uncertainty estimation, targeted data acquisition, and update rules need to be co-designed (Ross et al., 2011; Parisi et al., 2019; Kunze et al., 2018). Specialists need mechanisms to detect change points, separate stable traits from transient contexts, and preserve rare user-specific norms that are unacceptable to violate repeatedly.

Concretely, continual learning for local environments should evolve beyond preventing average-case forgetting on benchmark task sequences (Kirkpatrick et al., 2017; Parisi et al., 2019; De Lange et al., 2021). It should include: *locality-aware updates* that change only the memories, adapters, or skills implicated by a correction; *uncertainty-aware data acquisition* that asks for help or enters observation mode when the model is unsure; *change-point detection* to separate durable preferences from temporary exceptions; *privacy-preserving storage and aggregation* so sensitive histories need not be centralized; and *safety-constrained evaluation* that measures whether online improvement occurs without increasing catastrophic errors or eroding appropriate reliance (Lee & See, 2004; Hoff & Bashir, 2015; Amershi et al., 2019). These requirements do not replace standard continual-learning objectives; they specialize them to the assistive AI setting.

## 8. Research Challenge II: Fast Adaptation with Limited Data

Specialist models must adapt quickly because repeated mistakes are costly: a web agent should infer action preferences from a handful of accept/reject signals, and a household robot should learn the affordances of a cabinet after only a few attempts or demonstrations. Wearables raise the bar further: in-context action prediction should become useful within minutes of a routine, such as cooking, repair, or packing, using only lightweight feedback and without requiring explicit labels. Fast adaptation is therefore both a usability requirement and a safety requirement, since it reduces friction while limiting repeated harmful or high-stakes errors. This connects to few-shot learning, meta-learning, meta-reinforcement learning, imitation learning, inverse reinforcement learning, and preference learning (Finn et al., 2017; Duan et al., 2017; Rakelly et al., 2019; Ross et al., 2011; Ng & Russell, 2000; Ziebart et al., 2008; Christiano et al., 2017).

Fast adaptation is challenging because real-world supervision is sparse and ambiguous. In long-horizon tasks, an error could arise from perception, state estimation, planning, control, or an incorrect model of user preference, and user feedback will rarely isolate the cause. In home robotics, valuable supervision is often observational rather than didactic: robots must acquire new behaviors from one-shot

or few-shot visual imitation, extracting goals, constraints, object affordances, and motion cues without explicit rewards (Ross et al., 2011; Ng & Russell, 2000; Ziebart et al., 2008). In wearables, the same issue appears as latent intent inference from partial egocentric streams: the assistant must infer whether the user is searching, cooking, repairing, hesitating, or about to make a mistake before explicit correction is available (Bao et al., 2025; Engel et al., 2023; Guzov et al., 2025). In web agents, edits and rejected suggestions provide weak supervision over source preferences, autonomy boundaries, and acceptable risk, making preference learning directly relevant (Christiano et al., 2017; Amershi et al., 2019).

This pushes specialist learning toward representations that make few-shot generalization efficient and credit assignment structured. A fast learner must infer latent intent and preferences from minimal evidence, distinguish long-term behavioral patterns from transient context, and update the right component without destabilizing the rest of the system. For a home robot, this might mean updating an object affordance model or local map rather than the entire policy; for a wearable assistant, updating a routine-specific memory or user-state estimate; and for a web agent, changing a preference model over sources or formatting rather than its general browsing capability. Thus, fast adaptation is not merely rapid fine-tuning; it is selecting the lowest-risk adaptation mechanism that can solve the local problem while preserving the generalist prior and avoiding interference with previously learned behavior (Kirkpatrick et al., 2017; Parisi et al., 2019; De Lange et al., 2021).

## 9. Solution Pathways: *Observational* and *Organic* Data for Continual Specialization in-the-wild

A central implication of our position is that we should emphasize learning from observational and organic data streams rather than relying primarily on intentionally scaled domain-specific datasets. In the specialist regime, the most valuable data is what naturally occurs in use: browsing traces, edits, corrections, household interactions, and the continuous sensory stream of an embodied agent. This data is "organic" in the sense that it is produced as a byproduct of achieving goals, not as a curated dataset designed to train a model. The research challenge is to convert this stream into learning signal without requiring costly annotation or intrusive user burden.

For web agents, organic data includes the full action trace of browsing: which links were chosen, which content was extracted, which suggestions were accepted, and which were rejected. It also includes user edits that implicitly label what mattered. A specialist agent can treat these traces as preference supervision, learning a reward model that predicts user

satisfaction and then optimizing behavior accordingly. This approach reframes personalization as a continual reinforcement learning problem under implicit feedback, where the core difficulty is credit assignment across long sequences and the prevention of drift.

For home robots, organic data includes observation of human behavior as well as the robot's own trial-and-error. Observational learning becomes especially important because it offers a way to learn household norms without requiring exhaustive demonstrations. Humans routinely learn by watching others, inferring intent and structure rather than copying trajectories verbatim (Bandura, 1977; Meltzoff, 1995). A household robot can analogously learn from in-home observations of how objects are arranged, how tasks are performed, and what constitutes completion, using these observations to build goal detectors, progress estimators, and priors over acceptable actions. The robot can then refine through self-practice, using its own interactions to align perception with action.

This emphasis on observational learning is also a critique of a common trend in embodied AI: to solve specialization by intentionally scaling domain-specific datasets. That approach may deliver initial gains, but it risks a generalization wall because it cannot keep pace with the diversity of household-specific or user-specific conventions. Organic streaming data is, by construction, aligned with the local distribution. The challenge is not only curating this data, but designing methods that can transform noisy, biased, privacy-sensitive interaction traces into safe learning signal.

## 10. A Cognitive Science Perspective: Intuition from Human Adaptation

Human adaptation offers a useful analogy for the specialist view. Human competence is neither uniformly broad nor uniformly reliable across contexts; instead it is often *situated*, *goal-directed*, and shaped by the environments and communities in which people repeatedly act (Suchman, 1987; Hutchins, 1995; Lave & Wenger, 1991). A useful lens from cognitive science is that intelligence functions as an adaptive control system, continually adjusting perception, memory, and action to the rules and patterns of the world. Our effectiveness emerges through experience-based structure: home routines, workplace conventions, and personal preferences that encode operational constraints. In this sense, humans are prototypical specialists: we generalize strongly *within* familiar contexts, while transfer across contexts is often mediated by adaptation and fast construction of new local models.

A central theme is that cognition is *embodied* and *extended*: many capabilities emerge from tight loops between brain, body, and environment rather than from isolated internal

computation (Clark & Chalmers, 1998; Clark, 2008; Gibson, 2014). People offload structure into the world (notes, calendars, labeled drawers, habitual object placement), so intelligence is distributed across internal representations and external scaffolds. This directly mirrors specialist agents and robots whose competence is inseparable from persistent interaction with a particular environment: a household robot that learns which drawer sticks and which objects are fragile is not merely storing facts, but exploiting an *extended* representation coupled to stable regularities in the home. Likewise, a web agent that learns a user's trusted sources and workflows is building an externalized cognitive routine, analogous to how humans learn institutional procedures through repeated interaction.

Human learning is also *continual* and shaped by temporally correlated streams rather than i.i.d. samples. Research on statistical learning demonstrates that people learn patterns from unmarked data streams that lack explicit labels (Saffran et al., 1996; Aslin, 2017). Related work on intuitive psychology and causal learning suggests that people infer hidden structure—including goals, causes, and social norms—from sparse observations. The two models of inverse-planning and rational-action explain this process by showing how observers use constraints to understand goal-directed behavior which allows them to make quick intent predictions from small amounts of evidence (Baker et al., 2009). The specialist regime becomes more similar to social learning because people learn most of their skills through direct observation of others (Bandura, 1977). People who learn through observation tend to focus on achieving desired results instead of mimicking actions and they select information to replicate based on their understanding of cause-and-effect relationships (Meltzoff, 1995; Gopnik et al., 1999). This suggests a path for specialists—especially home robots—where organic observation and light-touch interaction provide the main supervision, rather than exhaustive labels or dense rewards.

Finally, cognitive science emphasizes the balance between *stability and plasticity*. People preserve habits that work, revise them under drift, and rely on multiple interacting memory systems—episodic traces, semantic abstractions, and procedural skills—with different update rates. This provides an analogy for a specialist system: a broadly competent backbone paired with fast-changing components like memory, routines, and preference models, along with mechanisms for change-point detection. Such a system can *specialize through lifelong interaction* by learning from organic streams, inferring latent preferences and norms, and improving safely under feedback loops and drift.

## 11. Alternative Views

One counterargument is that we can get most of the benefits of local specialization by pushing generalists far enough. If data from many users, including wearable and household traces, could be pooled and used to continuously update a single large model, then increasingly capable generalists might become "almost-specialists" for everyone. This is possible in principle, and our position is not a proof of impossibility. The narrower claim is that, in assistive deployment, broad priors must still be converted into reliable behavior for one local context, while the most informative local streams are often too privacy-sensitive, high-bandwidth, and non-stationary to treat as ordinary centralized training data. Personal adaptation may become a small last-mile problem, but that last mile still determines usefulness.

Concurrent work has similarly questioned AGI-style generality as the central objective, arguing that specialization and rapid adaptation may provide a more coherent target for AI progress (Goldfeder et al., 2026). Our paper emphasizes a complementary long-tail problem: assistive AI is not only about broad static task coverage, but about the last-mile challenge of adapting to the messy, longitudinal, and local structure of a particular user, environment, and interaction history after deployment.

Another counterargument is that generalist models are economically and scientifically superior because they amortize training cost and enable broad reuse. In this view, specialization is a thin layer atop a powerful base model, and the correct strategy is to push generality as far as possible and treat personalization as a product detail. This is plausible when specialization can be achieved cheaply and safely via prompting, retrieval, memory, or lightweight adaptation, and it describes a meaningful portion of current product progress. The challenge is that many specialist requirements are not thin: continual learning, preference tracking, and safe adaptation in embodied settings require algorithmic mechanisms and evaluation regimes that go beyond prompt engineering and minimal post-training.

A related concern is that specialization risks a fragmented ecosystem of models, each overfitting to a user and failing to generalize. This concern is real if specialization is achieved by uncontrolled fine-tuning that destroys transferable structure. However, the specialist thesis does not imply abandoning shared representations; it changes the *target* of optimization. We can still leverage shared backbones while insisting that the deployed artifact becomes specialized, and view specialization as a stress test: can a system adapt quickly without compromising compliance or requiring massive new data?

Finally, local specialization will often require very large generalist pretraining: without broad world knowledge, small-

data adaptation may be ineffective. This is correct in many settings, and aligns with the view that generalists are scaffolds. The disagreement is about emphasis. Treating generalist scaling as the main objective risks neglecting the adaptation mechanisms that convert broad priors into reliable local competence. This paper's position is to treat pre-training as acquisition of transferable representations and heuristics, and specialization as the central learning problem that determines real-world utility.

## 12. Implications for Research and Evaluation

A specialist-centric research program requires new evaluation regimes. By *static evaluation*, we mean a fixed dataset or environment snapshot evaluated once with a small set of metrics, often summarized as a single aggregate number. Such benchmarks reward broad competence, but they rarely measure whether a system improves over time with a user, learns preferences without explicit labels, or adapts safely in a local environment (Liang et al., 2022; Hutchinson et al., 2022). For web agents, evaluation should stress longitudinal interaction and interface drift; for home robots, it should stress within-home improvement, household-specific routines, and learning from failures without causing harm.

One concrete benchmark template is an *adaptation curve*: systems start with the same initial information in a fixed local environment and are scored not only by initial success, but by improvement after a bounded number of interactions, corrections, or demonstrations. Useful metrics include area under the adaptation curve, number of user interventions, correction burden, rate of repeated mistakes, and catastrophic-error rate during learning. A second template is a *personalization-with-privacy* benchmark: each system receives private local histories for simulated or consented users, adapts locally, and is evaluated on held-out user-specific preferences without exposing raw histories to the evaluator. These templates make fair comparison possible without requiring that every user's private data become public.

This also implies a different relationship between data and algorithms. In the specialist regime, the model's data stream is endogenous and continuous, which means the learning algorithm and the interaction policy are inseparable. Data collection is no longer a pretraining step; it is a behavior of the system. This pushes us toward closed-loop learning paradigms, where exploration, uncertainty, and user feedback are integral parts of the learning process.

Finally, specialization requires tackling research challenges for privacy, steerability, and on-device learning. If the most valuable data is personal, then learning methods must operate under privacy constraints while still enabling adaptation and user correction. This may involve federated learning, lo-

cal fine-tuning with secure aggregation, or architectures that explicitly separate generic priors from personal memory.

## 13. Conclusion

We argued that reliable assistive AI should treat local specialization after deployment as a central objective. The most valuable assistants will be those that become tightly coupled to a user and a bounded environment, learning continuously from streaming, organic data and adapting to evolving preferences, routines, and constraints. Web agents, wearables, and home robots illustrate why this is not a niche requirement: in these domains, correctness is local, the long tail matters, and post-deployment interaction provides the most relevant learning signal.

Generalist foundation models are an important starting point, but they do not eliminate the specialist problem. The core challenges are continual learning from streaming data and fast adaptation from sparse, implicit feedback, under constraints of privacy, safety, and user burden. Observational learning and organic data streams offer a promising substrate for this kind of specialization, echoing how humans become competent through immersion and practice rather than exhaustive supervision.

A specialist-centric agenda forces the field to confront difficult questions about safety, auditing, privacy, and evaluation. These are not reasons to avoid specialization; they are reasons to study it. If we want AI that integrates into daily life, we should optimize not only for models that can attempt everything for everyone, but for locally adaptive systems that do the right things for someone, and keep getting better alongside them.

## Acknowledgements

H.B. thanks Jaemin Cho, Krishna Murthy, Ben Newman, Irmak Guzey, Anand Bhattad, Chuhan Chen, Tianmin Shu, and Mononito Goswami for insightful discussions and feedback on the paper.

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
