# OpenReview forum: "Position: Assistive AI requires Personalized Specialists, not Generalists"
_ICML.cc/2026/Position_Paper_Track — ICML 2026 Position Paper Track regular_

### Official Review · Reviewer_yqYk · 2026-03-12

**Significance:** 3
**Argument Clarity:** 3
**Rating:** 4
**Confidence:** 4

**Questions:**

1. The paper suggests we need new evaluation regimes to measure "within-home improvement" and "longitudinal interaction." Given that the core value of a specialist model lies in its alignment with local idiosyncrasies, could you provide a more concrete example of what a standardized "Specialist Benchmark" would look like? How can we achieve fair cross-system comparisons without violating privacy constraints?

2. The paper strongly advocates for continual learning from post-deployment "organic, on-policy streams." However, in physical interaction scenarios (such as home robots), how do you address the risk of physical damage or catastrophic failures during the cold-start phase caused by early exploration, before sufficient "organic data" has been collected?

3. Could you further clarify the boundary in your definition between "maintaining long-term memory (e.g., via RAG or ultra-long context windows)" and "updating model weights"? If a generic foundation model merely ingests and retains a user's historical interaction traces over the past few months via an extremely large context window—thereby exhibiting personalization—does it satisfy your definition of a "specialist model"? Or must the model possess non-static parameter updating capabilities?

**Alternative Views Section:**

Yes

**Compliance With Llm Reviewing Policy A Conservative:**

Affirmed.

**Discussion Potential:**

3

**Paper Summary:**

This position paper argue that to build AI capable of truly and reliably assisting humans in their daily activities, the AI community needs to shift away from the current paradigm of over-pursuing "generalist foundation models" and focus instead on developing "Personalized Specialists." The authors emphasize that a "specialist model" here does not mean a model capable of executing only a single or narrow task; rather, it refers to systems that are tightly coupled to a specific individual usr and their local environment, continuously learning and adapting post-deployment.

The paper substantiates this position through case studies in three specific domains: AI web agents, assistive wearables, and home robots. In these scenarios, the system often does not need universal correctness across all environment, but rather needs to align with highly personalized local preferences, privacy constraints, and safety boundaries. To achieve this goal, the paper point out that the research focus should shift toward continual learning from post-deployment streaming data, fast adaptation using sparse feedback, and extracting learning signals from organic observational data, rather than solely relying on scaling domain-specific datasets. Finally, the paper explores human learning mechanisms as "specialists" from a cognitive science perspective and addresses potential counterarguments.

**Position:**

Yes

**Position In Title:**

Yes

**Related Work:**

3

**Strengths And Weaknesses:**

Strengths:

1. This topic is of practical significance to the ICML community. As the capabilities of large models continue to expand, adapting model to individual users in real-world deployment (the "last mile" problem) is becoming a core bottleneck hindering technology adoption. The paper distinguishes between the "breadth of task coverage" and the "degree of coupling to a local environment," which is a valuable contribution that helps clarify the community's confusion regarding "generalists" versus "specialists."

2. The logical flow of the paper is good. By introducing three concrete case studies (web agents, wearables, and home robots), the paper grounds its abstract position. It challenges the currently dominant implicit assumption of achieving "zero-shot generalization" through scaling, advocating instead for treating post-deployment streaming data as the primary mechanism for knowledge acquisition, which is thought-provoking.

Weaknesses:

1. As a position paper, while exhaustive experimental benchmarks are not strictly required, the current argument relies almost entirely on logical deduction and rhetoric. The position would be enhanced by adding a proof-of-concept—for example, quantitatively demonstrating how a "generalist backbone + local continual learning" outperforms a "pure generalist model" on a specific localized task.

2. Although the paper diagnoses the pain points and proposes research directions like "continual learning" and "fast adaptation," the proposed solutions (such as utilizing "organic streaming data") remain largely abstract. I suggest further exploring the architectural blueprint of these solutions—for instance, detailing how to design a system with a "generalist backbone paired with local memory and safety monitors," or briefly outlining the formulation of this constrained optimization problem.

3. The paper points out that current static benchmarks are no longer suitable and calls for new evaluation regimes that measure "within-home improvement" and "longitudinal interaction." However, given that "personalization" is inherently difficult to standardize for cross-comparisons, the paper's practical guidance value for the community would be improved if it could propose 1-2 concrete, actionable metric design ideas or benchmark frameworks.

**Support:**

3

---

> ### Author Rebuttal · Authors · 2026-03-31
>
> Thank you for the careful and thoughtful review. We very much appreciate that you engaged the paper on its intended terms. We are especially glad that the distinction between the breadth of task coverage and the degree of coupling to a local environment came through clearly, since that is the paper’s central conceptual contribution. We also appreciate your recognition that the topic is practically important for the community, and that the three case studies helped ground the position in concrete settings rather than leaving it at an abstract level. More broadly, we are encouraged that you saw the paper as identifying a real “last mile” bottleneck in assistive AI: not only building broadly capable systems, but making them reliably useful for particular users in particular environments over time.
>
> On the concern that the proposed solutions remain somewhat abstract: we agree that the paper should say more about what the architecture of a personalized specialist could look like in practice. Our intention was not to prescribe a single design, but to argue for a class of systems that combine strong pre-trained priors with local adaptation mechanisms. We agree, however, that the paper would be more useful if it included a more explicit blueprint—e.g., a generalist backbone paired with structured local memory, user- and environment-specific latent state, safety monitors, and constrained update mechanisms that allow learning from streaming data without unconstrained behavioral drift. We will revise the relevant sections to make this design space more concrete.
>
> On the question of evaluation and benchmark design: we agree that this is one of the most important places where the paper should be more actionable. One concrete specialist-benchmark template we have in mind is to evaluate systems not only by initial zero-shot performance, but by their adaptation curve within a bounded interaction budget in a fixed local setting. For example, systems could be compared on: (1) initial competence, (2) improvement after a fixed number of interactions or corrections, (3) user-correction burden, and (4) safety or catastrophic-error rate during adaptation. The key idea is that all systems begin from the same initial information and are evaluated on both speed and safety of specialization, rather than only static final-task success. We agree the paper should present 1–2 such benchmark templates more explicitly, especially in ways that preserve privacy while still enabling fair cross-system comparison. We will add this to the final version.
>
> On your question about cold-start risk in physical settings: we agree this is one of the strongest objections to the specialist thesis, especially for home robotics. Our view is not that early deployment should involve unconstrained exploration, but that specialization should be built on top of strong priors and conservative adaptation mechanisms: uncertainty-aware intervention, restricted action spaces early in deployment, monitoring and rollback, human approval for high-risk actions, and, where appropriate, simulation or shadow-mode learning before autonomous execution. In that sense, safety constraints are not separate from the learning problem; they are part of the learning problem itself. We will revise the paper to make that more explicit.
>
> On the distinction between memory and weight updates: our intended definition is mechanism-agnostic. A system can qualify as a specialist even if it does not continuously update all model weights. What matters is that the deployed artifact is non-static and becomes meaningfully shaped by local history, constraints, and feedback over time. That could happen through targeted parameter updates, structured memory, retrieval over persistent history, learned latent variables, or hybrid mechanisms. So yes, a foundation model that persistently incorporates a user’s history and environment in a way that changes behavior over time would fall within our notion of specialization; full-parameter updating is not required. We will make this boundary clearer in the final version.
>
> Finally, we appreciate that your review helped identify exactly where the paper can become more concrete and useful to the community. We believe the paper’s contribution is stronger than a general call for personalization: it tries to reframe specialization as a deployment-time property and to argue that post-deployment adaptation should become a first-class research target for assistive AI. We will revise the paper to operationalize this message more clearly. We hope these clarifications help address the main concerns, and we would greatly appreciate an improved assessment of the paper.

---

> > ### Author Rebuttal · Reviewer_yqYk · 2026-04-02
> >
> > Thanks for the response and clarification.
> > I maintain my original positive rating of 4.

---

### Official Review · Reviewer_ysST · 2026-03-12

**Significance:** 2
**Argument Clarity:** 3
**Rating:** 2
**Confidence:** 4

**Questions:**

## Additional comments

C1. Interesting perspectives that relate to this work are:
- Kunze, L., Hawes, N., Duckett, T., Hanheide, M., & Krajník, T. (2018). Artificial intelligence for long-term robot autonomy: A survey. IEEE Robotics and Automation Letters, 3(4), 4023-4030.
- Silver, D., & Sutton, R. S. (2025). Welcome to the era of experience. Google AI, 1, 11.

C2. L234 ("the next sections formalize"): the following sections don't give a formal treatment, whereas perhaps they should (see also W1 above).

C3. Section 10 "The challenge really is about curating it in order to learn from it safely and efficiently": Surely the methods for this learning are also an important challenge.

C4. Section 12: the last argument appears to be the same as the first one in this section?

C5. The illustrations in the last "column" of Figure 2 do not add much?

C6. Typos, etc.
- L122 RHS: "airine", "But [...] but" earlier in paragraph
- L371: spacing before and after references; missing period

**Alternative Views Section:**

Yes

**Compliance With Llm Reviewing Policy A Conservative:**

Affirmed.

**Discussion Potential:**

2

**Final Justification:**

My final recommendation is Reject, same as the initial one. The authors agreed with the issues I raised, but I believe they require significantly more than a rebuttal window to address properly.

**Paper Summary:**

This paper considers the future of assistive AI technology. It argues that, in order to be useful for people in their daily lives, such technologies should have a large degree of personalization. In contrast to directions being currently pursued by the research community, which seeks to (pre-)train generalist models, the paper calls for agents that spend time to familiarise themselves with, and adapt in response to, the environment they are deployed in. The paper looks at a few examples in web agents, wearables, and robotics and discusses the challenges in continual learning and fast adaptation.

**Position:**

Yes

**Position In Title:**

Yes

**Related Work:**

1

**Strengths And Weaknesses:**

## Strengths
S1. The paper addresses a timely topic that examines how we might move beyond the current wave of web-scale pre-trained approaches.

S2. The argument, at a high level, is quite compelling and the writing of the paper is generally good quality.

## Weaknesses
W1. The paper is too shallow in its coverage of the literature in continual, online, and lifelong learning -- both in general and as it pertains to e.g. robotics and the other application areas. It is not possible to understand what the current state of the art is in these areas and what improvements need to be made in order to render this vision plausible. In my view, especially for a machine learning audience, sections 8 and 9 should grow to become a major part of the paper, and these topics should be explored in depth.

W2. There are quite a few places in the paper that make assertions about what users want, how they behave, etc. that should be supported by evidence and citations. For example, "what determines their usefulness is not generic browsing skills but alignment to the end user’s idiosyncratic preferences", "users also differ in when they want clarification vs. autonomy, and these preferences vary with context and stakes", "People preserve habits that work revise them under drift, and rely on multiple interacting memory systems", "Research on learning demonstrates that people discover hidden patterns which include goals and causes and social norms to achieve their goals", *etc.*. The literature on human-computer /  human-robot interaction, cognitive science, and experimental psychology will be relevant here. Together with W1, this contributes to the impression that the paper is shallow in its understanding and coverage of the literature; despite its central thesis being intuitively compelling.

**Support:**

2

---

> ### Author Rebuttal · Authors · 2026-03-31
>
> Thank you for the constructive review. We appreciate your assessment that the core thesis is timely and compelling, and we also appreciate the concerns raised for improving the paper. They point to places where the paper can be strengthened substantially through a clearer framing of related work and tighter support for several empirical claims, without changing the underlying position.
>
> On W1, we agree that the paper currently under-signals its relationship to continual, online, and lifelong learning. Our intent in Sections 8 and 9 was not to offer a survey, but to identify the research challenges that become central once deployment-coupled specialization is treated as the target for assistive AI: learning from temporally correlated on-policy streams, adapting under drift, handling weak and implicit feedback, and doing so safely in privacy-sensitive settings. That said, for an ICML audience, the draft should do more than gesture at those themes. We will revise those sections so that they more clearly situate the position relative to existing continual-learning and long-term autonomy literatures, explain what is already known with more detailed references, and identify the remaining gaps that matter specifically for assistive systems. We are aligned with the reviewer that the paper doesn't need a different argument, but that it needs to show more explicitly how its argument sits on top of, and redirects attention within, those existing areas.
>
> We found W2 especially useful because it identifies a second place where the paper needs to ground the intuition for arguments with related literature. Several statements about user preferences, assistance, habit, memory, and adaptation are key conceptual pieces in the paper, and we appreciate the reviewer's assessment that they are intuitively compelling: they are part of the reason we argue that correctness in assistive settings is often local, preference-coupled, and only partially specifiable in advance. We will support them more carefully with relevant HCI, HRI, cognitive science, and psychology literature to strengthen the paper. This is a meaningful improvement we can make within the current paper, and it would strengthen the key thesis rather than change it.
>
> Your additional comments are also well taken as they point to statements that can be made more accurate and more useful in the final version. We will cite and discuss the papers in C1 (they indeed are very relevant and would strengthen our paper), and will incorporate the changes in C2 as mentioned above. On C3, you are right that the challenge is not only the curation of organic data but the methods required to learn from it safely and efficiently; that methodological challenge is central to the agenda, and the revised paper will say so directly. On C4, the distinction between the first and last counterarguments in Section 12 should be sharpened so that one is clearly about whether sufficiently broad pooled data could reduce the need for explicit specialization, while the other is about whether large-scale pre-training may still remain necessary even if specialization is essential. We will make those distinctions cleaner in the revision.
>
> More broadly, we are committed to making the changes suggested by the reviewer in the final version, and we believe they would strengthen the paper while remaining within the scope of the present submission. We would greatly appreciate your consideration of a revised assessment.

---

> > ### Author Rebuttal · Reviewer_ysST · 2026-04-01
> >
> > Many thanks for engaging with my comments. I am glad that you have found them useful and are considering addressing them in a future version.
> >
> > However, my assessment is that this requires a scholarly effort that is well beyond what can be achieved in a short rebuttal window. It is important to do a meaningful dive into both the relevant ML and non-ML literatures as I have outlined. In my opinion, it is not only a matter of finding citations to support the existing arguments. Even though they are indeed intuitive, often times scientific truth is not.
> >
> > I do think the work has potential and I would encourage the authors to keep pursuing it. However, it is not ready for publication in its current form.

---

### Official Review · Reviewer_eipJ · 2026-03-13

**Significance:** 2
**Argument Clarity:** 3
**Rating:** 4
**Confidence:** 3

**Questions:**

N/A

**Alternative Views Section:**

Yes

**Compliance With Llm Reviewing Policy A Conservative:**

Affirmed.

**Discussion Potential:**

3

**Final Justification:**

I would like to keep my rating as the rebuttal resolves my concerns.

**Paper Summary:**

This position paper argues that the long-term target for assistive AI should not be increasingly capable generalist models, but personalized “specialists” that become tightly coupled to a particular user and local environment over time. The authors explicitly define specialization not as narrow task scope, but as post-deployment adaptation to a user’s preferences, routines, constraints, and environment. They motivate this thesis through three domains: web agents, wearable assistants, and home robots, all of which they characterize as settings where correctness is local, user-specific, and revealed mainly through streaming on-policy interaction after deployment. The paper then identifies two central research challenges—continual learning from streaming data and fast adaptation from sparse, implicit feedback—and argues that observational or “organic” data should be the main substrate for building such systems. It closes by linking this agenda to cognitive science and by calling for new evaluation regimes centered on longitudinal improvement, safety, and privacy rather than static zero-shot competence.

**Position:**

Yes

**Position In Title:**

Yes

**Related Work:**

3

**Strengths And Weaknesses:**

- Timely and well-motivated. The paper addresses a real tension in current ML: impressive progress from large generalist models versus the practical need for systems that work reliably for one user in one environment over time.

- Clear reframing of “specialist” versus “generalist.” One of the stronger aspects of the paper is that it does not reduce specialization to single-task behavior. Instead, it treats specialization as tight coupling to local history, preferences, constraints, and feedback.

- Good choice of case studies. The three domains complement each other well. Web agents highlight preference alignment and interface drift; wearables foreground temporally dense, privacy-sensitive multimodal streams; home robotics makes the safety and long-tail adaptation problem especially concrete. The examples make the position legible rather than purely philosophical.

**Support:**

3

---

> ### Author Rebuttal · Authors · 2026-03-31
>
> Thank you for the thoughtful and encouraging review. We very much appreciate that you engaged the paper on its intended terms, and we are especially glad that the central distinction came through clearly: by “specialist,” we do not mean a narrowly scoped system, but one that becomes increasingly well matched to a particular user, environment, and history through post-deployment adaptation. That reframing is the core of the paper, so it is reassuring that you found it both clear and well motivated.
>
> We also appreciate that you understood the paper as arguing not against generalist models, but for a shift in emphasis. Our intended claim is that broad generalist pre-training may be an important starting point, but for assistive AI the harder and more consequential problem is how to convert broad priors into reliable local competence over time. In many assistive settings, success is only revealed through situated use, repeated interaction, and user-specific feedback, which is why the paper focuses on streaming specialization, continual learning, and fast adaptation from weak or implicit signals after deployment.
>
> We are also grateful that you found the three case studies effective. A major goal of the paper was to make the position concrete rather than purely philosophical, and to show that the same underlying issue appears across web agents, wearable assistants, and home robotics in different but complementary ways.
>
> Consistent with feedback from other reviewers, we plan to strengthen the paper in at least two ways: first, by grounding Sections 8–10 more deeply in the continual learning, online adaptation, and personalization literature; and second, by making the evaluation implications in Section 13 more concrete, especially around longitudinal improvement, safety under continued learning, and privacy-preserving personalization.
>
> We appreciate your supportive assessment, and are encouraged by the review. We hope these planned revisions help strengthen the case for the paper’s significance, and we would be very grateful if you would consider strongly supporting the paper.

---

> > ### Author Rebuttal · Reviewer_eipJ · 2026-04-02
> >
> > Thanks for the rebuttal. I'll maintain my original positive rating.

---

### Official Review · Reviewer_DD1s · 2026-03-14

**Significance:** 3
**Argument Clarity:** 2
**Rating:** 5
**Confidence:** 4

**Questions:**

As stated in my assessment, I think that this position paper summarizes at high level a sensible position. However, it fails to ground such high-level views into specific and novel low-level details and insights. More importantly, the reasoning is insufficiently strong to argue against generalist models and their possible evolution in the next years.

**Alternative Views Section:**

Yes

**Compliance With Llm Reviewing Policy A Conservative:**

Affirmed.

**Discussion Potential:**

3

**Final Justification:**

I updated my score, as the rebuttal changed my mind in several aspects.

I still think that the relation between the so-called specialists and existing topics within machine learning could be better explored and elaborated in the position paper.

However, the definition of personalized specialists carries certain novelty and relevance, and it is well supported, even if only by high-level arguments. I think the paper deserves to accepted for these reasons.

**Paper Summary:**

The position in this paper is that assistive AI requires specialist models and not generalist ones. Specifically, the authors constraint the meaning of specialist to the deployment domain and not to other dimensions, to differentiate their view from narrowing down the focus of the tasks in AI models, in contrast to other views in similar discussions. As further differentiation, the authors argue that specialization should be acquired at deployment, again in contrast to narrowing down the model goals. The paper particularizes the reasoning to 3 specific use cases: AI web agents, wearables and home robots, and identifies continual learning and adaptation as the main challenges.

**Position:**

Yes

**Position In Title:**

Yes

**Related Work:**

2

**Strengths And Weaknesses:**

While the paper's reasoning is sensible, in my opinion, it has serious weaknesses:

1) The paper's topics have been commonplace in the latest years in the machine learning community. While the paper reflects some of the motivations and reasoning that are typically heard, it does not articulate novel components or provide additional insights on the discussion. The authors propose research on continual learning starting from foundation models, or privacy-ensuring frameworks, which are already stablished research topics in the community.

2) The authors make the analogy with humans in Section 11. However, while biology may serve sometimes as an inspiration, it cannot be used to argue about the impossibility of a technology operating in certain manner. In many occasions, technology may differ substantially from biologic systems, and outperform them.

3) The arguments in the paper are not fully convincing on the impossibility of a generalist AI framework capable of succeding zero-shot in a wide variety of scenarios. Privacy issues, that are argued by the authors to motivate weak performance in home environments, may be solved by privacy-aware methods. The paper does not sufficiently analyze current trends in the ML community that seem to oppose to their view: generalist models have been improving at a fast pace in the last years, and they have outperformed specialist models in most tasks.

**Support:**

3

---

> ### Author Rebuttal · Authors · 2026-03-31
>
> Thank you for your thoughtful review. Your review usefully highlights a place where the paper needs to be more precise about what its contribution is, and is not. The paper is not intended to present continual learning, privacy-aware learning, or post-training adaptation as individually new research topics. Rather, its claim is that for assistive AI, these ingredients should no longer be treated as secondary “last-mile” additions to a generalist-first agenda, but as part of the primary scientific objective. The conceptual contribution we intended to make is therefore a reframing: “specialist” is defined not by narrow task scope, but by tight coupling to a user and local environment, with value arising from post-deployment adaptation. We will make that distinction much more explicit early in the paper so that the contribution is not misconstrued as a call for existing subareas and accurately reflects the substance of the position: a statement about what the field should optimize for.
>
> Relatedly, based on the reviewer's comments, we see that the current draft leaves too much room for the interpretation that the paper is arguing against the possibility of increasingly capable generalist systems. That is stronger than our intended position. Our intended claim is narrower and more practical: in assistive settings such as web agents, wearables, and home robots, the dominant determinants of reliability are often local preferences, local norms, and local non-stationarity, so broad pre-training alone is unlikely to be the end state of the most valuable deployed system. This is also why the paper describes a generalist-to-specialist pipeline rather than a rejection of generalist pre-training. To clarify this, we will re-phrase the writing so that the paper reads less as a forecast about the limits of generalists in the abstract, and more as an argument about the form that high-value assistive systems will likely take in deployment.
>
> The comment about current trends in ML is helpful, because it points to a discussion that we intend to make more explicit in the revision. The paper should engage more directly with the fact that generalist models have improved rapidly and often outperform narrow specialists on static tasks. Our point is not to deny that trend, but to argue that success on pooled-data, static, zero-shot evaluation does not resolve the hardest part of assistive AI: becoming reliable for one user, in one environment, over time, under privacy, safety, and feedback constraints. We can strengthen the paper by stating this relation more cleanly: generalist progress provides increasingly strong priors, but the unresolved problem for assistive deployment is converting those priors into stable local competence without repeated costly mistakes.
>
> On the cognitive-science discussion, we appreciate the chance to clarify the role that section should play. We agree that biology is not evidence of technological impossibility, and our intention was not to imply as though human specialization is being used to bound what AI can do. The intended role of that discussion is more modest: to motivate the idea that robust competence often emerges through repeated adaptation within stable local contexts, and that this makes specialization a natural lens for assistive systems. We can rewrite that section so it is clearly positioned as intuition and synthesis, not as an argument from biological limitation.
>
> We are committed to making these clarifications in the final version, since they sharpen the paper’s contribution without altering its central thesis. If these revision strategies address your main concerns, we would be very grateful if you would consider a revised assessment.

---

> > ### Author Rebuttal · Reviewer_DD1s · 2026-04-03
> >
> > I am grateful to the authors for their insightful rebuttal, I must admit that it changed my mind in several points, and hence I will upgrade my score accordingly.
> >
> > On my point 1, I see now novel components on how specialists are defined more regarding a local environment than specific tasks. I agree that this view carries certain novelty. I wonder, adaptation to local environments could be defined as a novel task instead of specialization?
> >
> > On my point 3, I agree with the authors that zero-shot and their view of specialists are different.
> >
> > As a matter of fact, "specialist" is a word that already carries a meaning and may confuse/bias readers... I encourage the authors, again, to give some thought to the possibility of coming up with a new flow for the paper that highlights differences from the beginning, maybe even renaming "specialist".
> >
> > On my point 2, if the authors tone down the text so that biological evidence is more clearly stated as a comment and not an argument of the position, it should be resolved.
> >
> > Still, I think the paper does not clearly point to what should be different about standard continual learning, privacy-aware learning, or post-training adaptation, in order to address local environments. How these techniques should be adapted or evolved to address this new task? I encourage the authors to elaborate on that in their revision of the paper.

---

> > > ### Author Response · Authors · 2026-04-07
> > >
> > > Thank you for the thoughtful follow-up and for revisiting your assessment. We are grateful that the rebuttal helped clarify the paper’s contribution.
> > >
> > > We also think your comment on the term “specialist” is well taken, and we agree the current wording may bring in some confusion from the outset. In the revision, we will change the framing to make the distinction clearer earlier in the paper, and we are open to replacing “specialist” with language such as locally adaptive assistive systems or deployment-adaptive assistive systems, which better emphasizes our intended meaning: systems whose value comes from becoming well matched to a particular user and environment after deployment, rather than from narrow task scope alone.
> > >
> > > Relatedly, on your question about whether adaptation to local environments could be framed as a novel task, we think that is a useful perspective, but we agree it is better for us to avoid terminology that may bias readers toward an unintended interpretation. Our intended point is broader than a single fixed task: assistive deployment requires ongoing adaptation to one user, one environment, and to shifting local norms over time. We will revise the paper so that this idea is introduced directly, without relying on potentially confusing inherited terminology.
> > >
> > > We also agree the paper should be more concrete about what is different for continual learning, privacy-aware learning, and post-training adaptation in this setting. In revision, we will make clearer that assistive deployment places these methods under a distinctive combination of constraints: privacy, sample efficiency, non-stationarity, and strong safety requirements because local mistakes are costly. We will sharpen this point in the final version. We also appreciate your note on the biology discussion and will revise that section so it is clearly presented as intuition rather than evidence. Thank you again for the constructive feedback and helpful suggestions for improving the paper.

---

### Decision · Program_Chairs · 2026-04-30

**Decision:**

Accept (regular)

**Comment:**

I acknowledge the discrepancies among the reviews, particularly the valid concerns raised regarding the depth of technical engagement with the continual learning literature. However, I think this paper can articulate a principled, thoughtful alternative to the status quo, irrespective of whether that alternative ultimately proves correct or is fully matured.

This paper provides a diverse and well-articulated counterpoint to the mainstream narrative of generalist scaling. While the community is heavily incentivized to pursue larger models and broader zero-shot benchmarks, this paper forces a necessary reflection on the "last mile" problem. It reframes the debate away from static task coverage toward the messy, longitudinal, and local nature of human assistance. This reframing, defining a "specialist" by its coupling to a local environment rather than by task narrowness, is a timely conceptual intervention that the ICML community needs to hear, even if the methodological roadmap is still under construction.

I nevertheless expect the authors to incorporate the literature and clarity improvements suggested by the reviewers in the final version.